# Effect of Dietary Forage/Concentrate Ratio on Nutrient Digestion and Energy and Protein Metabolism in Adult Donkeys

**DOI:** 10.3390/ani10061025

**Published:** 2020-06-12

**Authors:** Li-Lin Liu, Xiao-Ling Zhou, Hong-Jian Yang, Rong Chen

**Affiliations:** 1State Key Laboratory of Animal Nutrition, College of Animal Science and Technology, China Agricultural University, Beijing 100193, China; 120020004@taru.edu.cn; 2Key Laboratory of Tarim Animal Husbandry Science and Technology, College of Animal Science, Tarim University, Alar 843300, China; 120080009@taru.edu.cn (X.-L.Z.); 120020002@taru.edu.cn (R.C.)

**Keywords:** donkey, ration, fiber, crude protein, digestibility, metabolism

## Abstract

**Simple Summary:**

In recent decades, the donkey husbandry industry has developed rapidly, catering for their use in pharmaceutical, meat and milk production. However, compared with horses and other livestock, animal feeding studies have not been addressed to understand how nutrient digestion and metabolism are limited in donkeys. In this paper, the effect of the forage/concentrate ratio (F/C) in three experimental diets (low-fiber ration, medium-fiber ration and high-fiber ration) was investigated on N and energy balance using the total feces and urine collected method in a 3 × 3 Latin square experimental design. Decreasing the F/C significantly promoted protein digestibility and decreased fiber digestibility; increasing the F/C remarkably decreased N retention through the greater increase in N excretion in urine; decreasing the F/C linearly increased the conversion efficiency of digestible energy to metabolizable energy.

**Abstract:**

The domestic donkey is a unique equid species with specific nutritional requirements; however, limited feeding studies have been addressed so far to understand nutrient digestion and metabolism in donkeys. In the present study, six adult female Xinjiang donkeys (180 ± 10 kg live weight) were applied in a 3 × 3 Latin square design to investigate the effect of the forage/concentrate ratio (F/C) in three experimental diets on N and energy balance within 12 weeks. Rice straw and alfalfa hay were chosen as forage ingredients, and the diets included the following: (1) a high-fiber (HF) ration (F/C = 80:20), (2) a medium-fiber (MF) ration (F/C = 55:45), and (3) a low-fiber (LF) ration (35:45). After the fixed amount of diets were daily allowed to the animals, total feces and urine were collected to determine total tract digestibility, N and energy balance. As a result, dry matter intake did not differ among the three diet groups. Decreasing the dietary F/C significantly promoted protein digestibility and decreased fiber digestibility. The N and energy balance analysis showed that increasing the F/C remarkably (*p* < 0.01) decreased N retention through the increase in N excretion in urine, and the highest N loss relative to N intake was observed in MF. Meanwhile, decreasing the F/C linearly increased the conversion efficiency of digestible energy to metabolizable energy. Taken together, the results obtained in the present study implicated that the dietary forage level should not be less than 55% to maintain greater N and energy utilization in feeding practice, otherwise, a donkey’s N utilization might be highly discounted.

## 1. Introduction

Equines as herbivore animals can make good use of low-quality roughages by relying on their well-developed hindgut, and they can also digest some amounts of dietary proteins, fats, starches and other soluble carbohydrates through the activity of intestinal digestive enzymes [1,2,3]. The donkey, belonging to the Equidae family, has certain specific variations from the horse [4].

The donkey is less of a flight animal and has been used by humans for pack and draught work over time, in areas where their ability to survive poorer diets can be utilized. When living as a companion animal, however, the donkey easily accumulates adipose tissue, and this may create a metabolically compromised individual prone to diseases of excess such as laminitis and hyperlipemia [4]. 

An earlier study reported 39.5% crude fiber digestibility and 36.1% acid detergent fiber (ADF) digestibility of alfalfa hay in horses [5]. The respective values for alfalfa hay digestibility in donkeys are significantly higher, at 54.2% neutral detergent fiber (NDF) and 46.8% ADF [6], suggesting that the donkey is more efficient than the horse in digesting the cell wall constituents. A recent review noted that donkeys take advantage of the very high N-recycling ability, and their energy and protein requirements are much lower than those of other equids [7]. The domestic donkey is a unique equid species with specific nutritional requirements. To maintain good health and welfare, the National Research Council (2007) provided dry matter intake, ration formulation and nutrient allowance recommendations (e.g., digestible energy, protein, calcium and phosphorus) for donkeys fed on poor- or good-quality forage [6].

Donkeys have been farmed in China for more than 4000 years. Although the donkey population in China fluctuated quite a lot in the past decades due to illegal skin trading across the world, the rapid development of the skin-use ejiao industry has boomed the population up to 6 million in China. Except for the skin-use of donkeys in China, meat or milk production also attract great interest by the national government and local communities, particularly in Xinjiang of northwest China. In this case, more and more donkey producers are wondering if decreasing the dietary forage/concentrate ratio (F/C) could alter nutrient digestion and metabolism. In this study, rice straw and three diets with different proportions of rice straw and alfalfa hay were formulated to result in namely a medium-fiber diet (MF), a high-fiber diet (HF) and a low-fiber diet (LF). The objective was to investigate the effect of decreasing dietary fiber content on nutrient digestibility and metabolism adult donkeys.

## 2. Material and Methods

### 2.1. Experimental Design

The experiment was designed as a 3 × 3 Latin square, with data being recorded for six cecum-fistulated donkeys during three experimental periods, where each period consisted of 21 days of adaptation to the diet and five consecutive days of feces and urine collection, and with three dietary treatments being tested per period. The whole experiment was completed in 12 weeks. All the donkeys remained healthy throughout the study, and they were cared for according to the Chinese laws and regulations concerning experiments on live animals (approved by the State Council on 31 October 1988, and promulgated by Decree No. 2 of the State Science and Technology Commission on 14 November 1988, with a second revision on 18 July 2001; protocol AW23050202-1).

### 2.2. Animals and Management

The experiment was carried out from July to September 2017, at the Key Laboratory of the Tarim Animal Husbandry, Science and Technology Xinjiang Production and Construction Group, of Tarim University. Six nonlactating female crossbred Xinjiang donkeys, 5 to 9 years of age, and with initial body weights (BW) of 180 ± 10 kg, were used as the experimental animals. They were surgically fitted with permanent cecal fistulas (inner diameter Φ 40 mm, length 50 mm) located near the ileocecal junction. The donkeys were housed in 2 × 5 m individual stalls with sand-rich earth beddings, and allowed to take a daily walk in outside pens (10 × 20 m) with the same bedding at 10:00 for 1.5 h. Fifteen days prior to the start of the experiment, the donkeys were treated with ivermectin, both topically and as an anthelmintic. All donkeys had free access to drink water and the MF ration (Table 1) prior to the start of the experiment. Daily feed intake was measured to determine the fixed amount of experimental rations during the subsequent feeding trial.

### 2.3. Feedstuffs, Diets and Feeding

Three rations with different forage to concentrate ratios (F/C) were fed to the donkeys during the formal feeding trial. As shown in Table 1, the experimental rations included an LF ration (F/C = 35:65), an MF ration (F/C = 55:45) and a HF ration (F/C = 80:20). The forage consisted of rice straw and alfalfa hay, which were purchased from a local hay producer, and chopped into approximately 1 cm lengths. The feed ingredients in the concentrates were purchased from the Kashgar TECON Feed Co., Ltd., (Kashgar, China). The concentrates were prepared once a week. According to the feed intake measurement prior to the start of the experiment, 1.2 daily MF ration intake (in kg) was offered into three equal meals and given at 8:30, 13:30 and 19:00 after the forage was wet with 30% water. To determine the feed intake of each animal, leftovers the next morning were recorded and weighed during each 21-day Latin square period, and all the animals had free access to a vitamin/mineral lick block during the whole animal feeding trial.

### 2.4. Sample Collection

During the 21-d experimental period, representative diet samples of each feeding treatment were collected. At the end of the experimental period, feces and urine of each animal were continuously collected and weighed daily for five days [8]. Prior to urine collection, 100 mL of 10% H_2_SO_4_ was added to the urine collection tank to avoid urinary N loss. Diet and feces samples were oven-dried at 65 °C for 48 h and then ground to pass through a 425 µm sample sieve. Resultant samples were sealed into plastic bags and stored at room temperature for subsequent feed analysis. Approximately 10% of the daily urine output per donkey was placed in a closed container at 4 °C and later stored at −20 °C for subsequent determination of the urine energy and N content.

### 2.5. Chemical Analyses

Representative samples of diet and feces were analyzed following the Association of Official Analytical Chemists (AOAC; [9]) for dry matter (DM, ID 973.18) and crude protein (CP, N × 6.25, ID 4.2.08). Following the method of Van Soest et al. [10], neutral detergent fiber (NDF) was determined with heat-stable amylase and sodium sulphite addition and expressed inclusive of residual ash, and acid detergent fiber (ADF) was determined and expressed inclusive of residual ash. The energy content of diet, feces and urine was determined using a bomb calorimeter (XRY-1B, Changji Geological Instrument Co., Ltd., Shanghai, China).

### 2.6. Calculation and Statistical Analyses

The apparent total tract digestibility coefficients of dietary DM, CP, NDF and ADF were calculated by the difference between daily ingested nutrient mass (g) and daily excreted nutrient (g) divided by daily ingested nutrient mass (g), and the values of coefficients were expressed in %.

Regarding the daily energy balance of donkeys, the digestible energy (DE) intake was calculated by subtracting the energy loss in the feces from the dietary gross energy (GE) intake. The metabolizable energy (ME) was calculated by subtracting energy losses in feces and urine from the dietary GE intake.

The retained N was calculated by subtracting the N excreted in the feces (FN) and urine (UN) from the dietary N intake (IN). Metabolizable protein intake (N × 6.25) was calculated by subtracting N losses in feces and urine from the dietary N intake. The biological value (BV) of dietary protein was calculated as follows:$$\mathrm{BV}=\frac{\mathrm{IN}\left(g\right)-\mathrm{FN}\left(g\right)-\mathrm{UN}\left(g\right)\;}{\mathrm{IN}\left(g\right)-\mathrm{FN}\left(g\right)}\times 100$$

To estimate the efficiency of digestible N converting into retained N, the N metabolic rate (NMR) was calculated as follows:$$\mathrm{NMR},\%=\frac{\mathrm{IN}\left(g\right)-\mathrm{FN}\left(g\right)-\mathrm{UN}\left(g\right)\;}{\mathrm{IN}\left(g\right)}\times 100$$

The data in the experiment were analyzed as a 3 × 3 Latin square design using the MIXED procedure in the SAS software (version 9.2; SAS Institute Inc., Cary, NC, USA). The model was applied as follows:*Y_ijk_* = *μ* + *T_i_* + *P_j_* + *C_k_* + *e_ijk_*
where *Y_ijk_* is the response variable, μ is the overall mean, *T_i_* is the fixed effect of the diet treatment (LF, MF, HF), *P_j_* is the random effect of the period (j = 1 to 3), *C_k_* is the random effect of the animal (k = 1 to 6) and *e_ijk_* is the residual error. The results are presented as means ± standard errors (S.E.), and the means were compared among the treatments using the Duncan method. Significance was declared at *p* < 0.05 unless otherwise noted.

## 3. Results

All the donkeys remained healthy throughout the trial.

### 3.1. Effect of Dietary Forage/Concentrate Ratio on Feed Intake and Total Digestive Tract Digestibility

As shown in Table 2, DMI numerically ranked as LF > MF > HF, though no significant difference occurred among the three diet treatments.

As shown in Table 3, increasing the dietary forage/concentrate ratio significantly decreased the digestibility coefficients of DM, CP and energy and increased the digestibility coefficients of NDF and ADF.

### 3.2. Effect of Dietary Forage/concentrate Ratio on N Metabolic Rate

As shown in Table 4, increasing the dietary F/C ratio significantly decreased the dietary N intake, but it did not alter N loss in feces. However, the LF group presented the highest N loss in urine while no differences occurred between the MF and HF groups. Consequently, the lowest N retention occurred in the HF group, but no difference was observed between the LF and MF groups.

Metabolizable protein intake was calculated by subtracting N losses in feces and urine from the dietary N intake, and the lowest intake occurred in the HF group, while no difference was observed between the LF and MF groups. As a result, BV ranked as MF > LF > HF, and the lowest NMR occurred in the HF group, while no difference was observed between LF and MF.

### 3.3. Energy Utilization

As shown in Table 5, increasing the dietary F/C ratio numerically decreased the feed energy intake, but it did not affect energy loss in urine. Increasing the dietary F/C ratio significantly increased energy loss in feces. Consequently, both digestible energy and metabolizable energy intakes were significantly decreased by the increase in the dietary F/C ratio.

## 4. Discussion

A large amount of research has focused on horse nutrition, and some studies have explored the differences in nutritional physiology between horses and donkeys [11,12]; however, a complete understanding of the digestive physiology of donkeys is still lacking. The main purpose of this study was therefore to analyze the effects of the dietary F/C ratio on nutrient digestion, N metabolism and energy utilization in donkeys.

Rice straw, alfalfa hay, corn meal and soybean meal were selected as the main feed ingredients used to formulate the experimental diets in this study, as these four feed resources are abundant in Kashgar and have always been the main feed resources used for donkey diets in commercial production.

In this study, the fiber levels of the three treatment diets varied by varying the inclusion levels of the rice straw, and the corn meal content was varied to produce treatment diets with different starch contents. The alfalfa hay was mainly included to meet the donkeys’ requirements for soluble fiber and vitamins, and soybean meal was the main source of protein.

### 4.1. Feed Intake

In the present study, all donkeys prior to the formal experiment were fed with the MF ration to obtain the amount of feed allowance. Afterwards, the donkeys were provided 1.2 times the amount of the corresponding experimental diet to determine their daily feed intake during the last five days of each Latin square experimental period. Increasing the dietary F/C ratio numerically decreased DMI, but no significant difference occurred among the three diet treatments.

A previous study noted that daily DMI for the maintenance requirement was recommended as 1.3%–1.8% of live body weight [13]. It is important to provide enough fiber in the diet to maintain the activity of the cellulolytic bacteria in the cecum and to stabilize the cecal environment [14,15]. In addition, including an appropriate proportion of fiber also ensures the diet’s palatability. The fiber digestibility of the HF group was higher than that of the LF group in horses [16].

### 4.2. Total Tract Digestibility

In theory, fiber fermentation can occur throughout the digestive tract [17], but the cecum is the main site of fiber digestion in the donkey. The NDFd of the LF group was lower than that of the HF group (*p* < 0.01), as well as being lower than that of the MF group, although this difference was not significant. As the fiber content in the diet increased from 32.82% in LF to 45.09% in MF, it increased by 12.27%, and the NDFd increased by 72 g/kg. The fiber content in the diet increased from 45.09% in MF to 59.32% in HF, an increase of 14.23%, and the NDFd increased by 104 g/kg.

The different pattern of differences between the treatments was found for the ADFd as for the NDFd, with the LF group having a lower value than the HF group (*p* < 0.01), but a non-significantly higher value than the MF group. This effect on fiber digestion was related to the undesirable changes in the cecum ecosystem that result from the ingestion of a high-concentrate diet, as a deteriorated cecum environment inhibits the activity of the cellulolytic bacteria, thereby decreasing fiber digestibility [18].

Nonetheless, increases in the dietary fiber content did reduce the digestibility of the CPd. In this study, the NDF content was 59.32% for the HF group, 45.09% for the MF group and 32.82% for the LF group, and as the fiber level increased, the CPd decreased. J.-P. Jouany, et al. (2008) also found that DMd and CPd were greater (*p* < 0.001) in the high-starch than high-fiber diet in horse [16]. This is very similar to the results of this article.

Previous studies have established that the total digestive tract digestibility of starch is close to 100% [19]. An increase in the starch content of the diet therefore increases the digestibility of the DM and organic matter [20,21,22]. The starch contents of the three treatment diets in this study were 0.2 g/kg feed for the HF diet, 1.2 g/kg feed for the MF diet and 2.5 g/kg feed for the LF diet. This difference in the starch content provides some explanation for the higher full digestive tract digestibility found for the LF than the MF group, and for the MF than the HF group. It is discouraged to use grain-based high-concentration diets to feed donkeys, as this will increase the donkey’s suffering from gastric ulcers [23], laminitis and colic [24].

The DMd and CPd were both negatively correlated with the dietary fiber levels, whereas the NDFd and ADFd were positively correlated with the fiber content of the diet. N utilization may affect the efficiency of fiber digestion through microbial activity in the cecum and colon [25].

Some studies have also indicated that donkeys digest dietary fiber in the forms of acid detergent fiber (ADF) and neutral detergent fiber (NDF) more effectively than other equids [12,26,27]. A previous study comparing the digestive efficiencies of different equine species under specific dietary conditions found that roughage residues were retained for longer in the digestive tracts of donkeys than horses, suggesting that the digestion efficiency of fiber was higher in donkeys than in horses [26]. When given free access, the roughage intake of horses is higher than that of donkeys; however, the higher apparent nutrient digestibility in donkeys compensates for their lower intake [12].

Previously reported NDFd values for horses have ranged from 38% to 60% [28]. In this study, the NDFd of the donkeys ranged from 22.86% to 40.53%, with this value decreasing with a decrease in the proportion of roughage in the diet. Izraely H and Choshniak I (1989) reported that the NDFd and ADFd of alfalfa hay in donkeys could be as high as 54.2% and 46.8% [29].

### 4.3. Metabolic Rate of N

It is already known that the mechanism of nitrogen metabolism in equine animals works as follows: the ingested feed protein is partially digested by enzyme hydrolysis in the small intestine, and those proteins that are not digested in the small intestine are decomposed into amino acids and ammonia by the microorganisms in the hind intestine, and ammonia is used to synthesize microbial proteins [30,31]. Proteins are fermented in the large intestine to produce ammonia, which is used by large intestinal microorganisms to synthesize amino acids [32]. Although these amino acids cannot be effectively used by horses, they are essential for the growth and maintenance of the large intestinal microflora, which are necessary for the fermentation of the dietary fiber population [32,33].

In the pony, nitrogen efficiency, calculated by Hintz et al. (2008) as absorbed N–(urinary N–endogenous N)/absorbed N, is 45% or 25% to maintain zero N balance at maintenance when fed non-urea (soya or linseed meal) or urea-supplemented diets, respectively [34]. These nitrogen balance-related indicators are to yet be established in the donkey. When donkeys are fed low-protein roughages such as wheat straw, donkeys use nitrogen more efficiently than ponies [29,35]. Vermorel et al. (1997) reported that high fecal N excretion rates indicated that the protein digestibility of the ingested diet was low in sport horses [36], and it was found in this study that the two high-fiber diets had the highest excretion rates of fecal N. A recent review noted that donkeys take advantage of their very high N-recycling ability, and their energy and protein requirements are much lower than those of others [6].

### 4.4. Utilization of Energy

The concentration of GE intake of the three treatment groups is not significant, but the energy digestibility is different (*p* < 0.01). As the F/C ratio increases, the dietary fiber level increases and the ADE decreases, as the fiber content in the diet increased 12.27% from LF to MF and the ADE decreased by 9.5%, and the DE obtained from each unit kilogram of feed decreased by 1.6 MJ/kg. The same fiber content in the diet increased 14.23% from MF to HF, ADE decreased by 5.7% and DE decreased by 0.8 MJ/kg. The metabolic rate of energy also decreased, while the loss of FE and UE increased. The main reason for this is that the increase in fiber and decrease in protein in the diet caused a decrease in energy use and increased losses [37,38].

The digestion of soluble nutrients in the diet is higher in donkeys than in ruminants [29]. Donkeys can effectively use soluble carbohydrates and proteins because these nutrients are mainly absorbed in the small intestine, and therefore avoid the losses caused by fermentation in the hindgut. This enables donkeys to obtain more energy from soluble nutrients than ruminants do [39].

Equines typically have a limited ability to digest roughages, but this is compensated for by their feeding behavior. Horses eat for 14 to 16 h per day [40], whereas donkeys eat for 14 to 17 h per day, to satisfy their requirements for roughage intake [41]. In their wild state, donkeys have to range over long distances (20–30 km/d) and spend longer time foraging, up to 14–18 h per day [42]. However, under house feeding conditions, donkeys do not need to travel long distances to forage, and feed processing and preparation are more scientific, therefore, the energy and time consumed for foraging should be significantly reduced, but there are no reports in this regard.

The energy utilization results were consistent with the digestive properties of the feeds and the digestive physiology of donkeys. Therefore, the low-fiber diet had the highest DE value, as well as the highest ME value, as the ME is largely dependent on the DE.

Studies have shown that the utilization efficiency of the GE of roughage, as indicated by the DE and ME values, is lower than that of concentrates and mixed diets [43].

## 5. Conclusions

The present study indicated decreasing the dietary forage/concentrate ratio significantly promoted protein digestibility and decreased fiber digestibility. The N and energy balance analysis showed that increasing the F/C remarkably decreased N retention through the greater increase in N excretion in urine, especially for the high F/C diet, and decreasing the F/C linearly increased the conversion efficiency of digestible energy to metabolizable energy. Taken together, the results obtained in the present study suggested that the dietary forage level should not be less than 55% to maintain greater N and energy utilization in practical feeding.

## Figures and Tables

**Table 1 animals-10-01025-t001:** Feed ingredient and nutrient level of low- (LF, forage/concentrate ratios (F/C) = 35:65), medium- (MF, F/C = 55:45) and high-fiber (HF, F/C = 80:20) donkey rations with different forage/concentrate ratios (F/C).

Items	LF	MF	HF
Feed ingredient, g/kg dry matter basis			
rice straw (1.0 cm)	250	450	700
alfalfa hay (1.0 cm)	100	100	100
cracked corn	352	150	0
soybean meal	56	50	25
cottonseed meal	217	225	150
premix feed	25	25	25
Nutrient level on dry matter basis			
Gross energy, MJ/kg	17.8	17.5	17.0
crude protein, g/kg	178	168	122
neutral detergent fiber, g/kg	328	451	593
acid detergent fiber, g/kg	238	346	469

**Table 2 animals-10-01025-t002:** Feed intake in adult donkeys fed low- (LF, F/C = 35:65), medium- (MF, F/C = 55:45) and high-fiber (HF, F/C = 80:20) diets with different forage/concentrate ratios (F/C).

Items	LF	MF	HF	*p*-Value
Dry matter intake, kg/d	4.17 ± 0.52	4.04 ± 0.74	3.77 ± 0.83	0.33

**Table 3 animals-10-01025-t003:** Total digestive tract nutrient digestibility coefficients of low- (LF, F/C = 35:65), medium- (MF, F/C = 55:45) and high-fiber (HF, F/C = 80:20) diets fed to adult donkeys with different forage/concentrate ratios (F/C).

Items	LF	MF	HF	*p*-Value
DM, %	62.3 ± 3.6 ^a^	51.7 ± 2.5 ^b^	45.2 ± 2.3 ^c^	<0.01
CP, %	84.5 ± 1.8 ^a^	80.9 ± 2.8 ^ab^	78.6 ± 2.7 ^b^	0.01
NDF, %	22.9 ± 2.3 ^b^	30.1 ± 2.6 ^ab^	40.5 ± 2.5 ^a^	<0.01
ADF, %	20.6 ± 2.4 ^b^	14.5 ± 2.1 ^b^	31.2 ± 3.2 ^a^	0.01
energy, %	69.0 ± 1.7 ^a^	59.6 ± 1.6 ^b^	54.2 ± 1.2 ^c^	<0.01

^a–c^ Means within a row with different superscripts differ at *p* < 0.05. DM, dry matter; CP, crude protein; NDF, neutral detergent fiber; ADF, acid detergent fiber.

**Table 4 animals-10-01025-t004:** Nitrogen balance in adult donkeys fed low- (LF, F/C = 35:65), medium- (MF, F/C = 55:45) and high-fiber (HF, F/C = 80:20) diets with different forage/concentrate ratio (F/C).

Items	LF	MF	HF	*p*-Value
Feed N intake, g/d	122.7 ± 15.89 ^a^	116.2 ± 21.31 ^a^	81.4 ± 18.07 ^b^	<0.01
Fecal N loss, g/d	21.2 ± 4.23	24.0 ± 4.03	19.1 ± 5.28	0.07
Urinary N loss, g/d	51.9 ± 5.33 ^a^	41.0 ± 3.21 ^b^	42.5 ± 4.34 ^b^	0.05
N retention, g/d	49.6 ± 12.68 ^a^	51.1 ± 15.99 ^a^	19.9 ± 8.56 ^b^	<0.01
Metabolizable protein intake, g/d	310.3 ± 79.27 ^a^	319.4 ± 99.98 ^a^	124.6 ± 53.55 ^b^	<0.01
BV (%)	48.3 ± 7.25 ^ab^	55.4 ± 10.74 ^a^	35.6 ± 22.74 ^b^	0.03
NMR (%)	40.1 ± 6.49 ^a^	43.9 ± 9.52 ^a^	27.3 ± 7.45 ^b^	0.03

^a,b^ Means within a row with different superscripts differ at *p* < 0.05. BV, biological value of protein; NMR, nitrogen metabolic rate.

**Table 5 animals-10-01025-t005:** Energy balance in adult donkeys fed low- (LF, F/C = 35:65), medium- (MF, F/C = 55:45) and high-fiber (HF, F/C = 80:20) diets with different forage/concentrate ratio (F/C).

Items	LF	MF	HF	*p*-Value
Feed energy intake, MJ/d	74.0 ± 3.3	70.6 ± 5.4	64.2 ± 5.9	0.12
Fecal energy loss, MJ/d	22.2 ± 1.4 ^b^	27.8 ± 2.2 ^ab^	29.0 ± 2.8 ^a^	0.03
Urine energy loss, MJ/d	20.4 ± 4.0	19.3 ± 3.7	20.0 ± 4.9	0.96
Digestible energy, MJ/d	51.2 ± 2.9 ^a^	42.0 ± 3.4 ^b^	34.7 ± 3.1 ^c^	<0.01
Metabolizable energy, MJ/d	30.8 ± 33.9 ^a^	22.7 ± 4.5 ^ab^	14.7 ± 2.2 ^b^	0.02
k = ME/DE	0.61 ± 0.07	0.53 ± 0.08	0.47 ± 0.10	0.39
q = ME/GE	0.42 ± 0.05	0.31 ± 0.05	0.25 ± 0.06	0.08

^a–c^ Means within a row with different superscripts differ at *p* < 0.05. DE, digestible energy; ME, metabolizable energy; k, coefficients of DE converting to ME; q, coefficients of feed energy converting to ME.

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
