# Peer review of "Effect of Dietary Forage/Concentrate Ratio on Nutrient Digestion and Energy and Protein Metabolism in Adult Donkeys"

_animals, 2020, doi:10.3390/ani10061025_

Round 1
Reviewer 1 Report
The manuscript has been improved a lot due to the revisions. However, there are some details that need to be revised:
- you have to revise the references in the text to be according to the instructions (e.g. initials are not used; all references are presented as numbers, also after the author names: e.g. you can write that "Author et al. [1] concluded that ..."
- the terminology still needs to be checked in English writing
- you have to tell the interval of feed sample collection in the text, not only to say "representative samples"
Author Response
- you have to revise the references in the text to be according to the instructions (e.g. initials are not used; all references are presented as numbers, also after the author names: e.g. you can write that "Author et al. [1] concluded that ..."
- Response 1:Yes, accepted and revised.
- the terminology still needs to be checked in English writing
- Response 2:
- you have to tell the interval of feed sample collection in the text, not only to say "representative samples"
- Response 3:Yes, accepted and revised.
Reviewer 2 Report
The manuscript is well written. It improves the knowledge for the management of donkey. Before its pubblication on Animals needs minor revisions.
All abbreviations in the main text and highlights should be given per extention at the first time of mention
Author Response
All abbreviations in the main text and highlights should be given per extention at the first time of mention
Response: Yes, accepted and revised
This manuscript is a resubmission of an earlier submission. The following is a list of the peer review reports and author responses from that submission.
Round 1
Reviewer 1 Report
Thank you for this submission. Clearly a very large scale study with a lot of work has been done and the methodology overall is sound. This is indeed valuable to present and publish but must be explained, discussed and evaluated more in the context of previous research and understanding. Some key research on donkey nutrition has not been integrated and justification for feeding high sugar starch feed is not clearly made. The results very much follow similar patterns of understanding in equine species overall but this is not clearly explained / understanding of this not really shown. So a valid piece of research which could be written up much better and clearer.
I have attached my comments together with a list of references.

Author Response
Response to Reviewer 1 Comments
Introduction
Thank you for your detailed review of my paper, you let me see the deficiencies of my article. Your careful and professional revisions have improved the quality of my articles, and I have learned a lot from them. There may still be many shortcomings, please criticize and correct me.
The following is my point-to-point response to your comments.
Point 1: 58 Horses cannot digest large amount of soluble carbohydrates or fats – please amend to ‘some’ amounts
Response 1: Yes, accepted and revised.
Point 2: 66. ‘ranked first place in the world’ – what does that mean? Rephrase – do they have the largest population?
Response 2: Yes, it means the largest population. Now we have rephrased the sentence.
Point 3: 68 this is totally out of date – fermentation is fermentation – but indeed comparison between donkey and goats have been made in the past and horses – need to work this up properly – see NRC 2007 chapter……. Also refer to current knowledge microbiome here and comparison between equine and ruminant – then go on to studies in donkeys
Response 3: The description has been deleted now.
Point 4: 75 In the last decades the donkey breeding industry….
Response 4: Yes, accepted and revised.
Point 5: 78 need to correct English --- high energy digestibility?? Not been mentioned before – too vague – why are we comparing to ruminant – what are the numbers then? For each – which ruminant are you comparing to? Too vague…
Response 5: The description has been deleted now.
Point 6: 81 , and there is no substitution…….onwards – this sentence makes no sense
Response 6: The description has been deleted now.
Point 7: 86 OUT OF DATE National Research Council 1986 – Did you mean that? NRC 2007 has a full chapter on donkey requirements with the following tables:
Comparative Energy Expenditures in Horses and Donkeys 271
13-2 Chemical Composition (g/100 ml) of Milk of Donkeys and Other Animal Species 273
13-3 Daily Rations for Adult Donkeys 273
13-4 Estimated Nutrient Intakes for Adult Donkeys Consuming Diets Based on
Poor or Good Quality Forage (dry matter basis) 273
Response 7: The description has been deleted now.
Point 8: 88 Again- limited review – previous studies need to be explored and summarised here (a lits of studies given at the end) – all too generic – any comparison of digestive physiology/volume of sections compared to horse?
Some repetition in intro and all too vague – not up to date – not building on previous research, knowledge and understanding – You need to clearly justify the study – are donkeys in danger of malnutrition – is this aimed at working donkeys and how much work do they do?
Response 8: The generic description has been deleted now, and the paragraph has been reworded now.
Methods
Point 9: 101 take out ‘repetitive’
Response 9: Yes, accepted and deleted.
Point 10: 105 sentence – During each….. I am not sure what you are trying to say here but I suspect this sentence is unnecessary – it is inherent in a latin square design?
Response 10: Yes, accepted and deleted.
Point 11: 118 please give: daily turnout times- what was on the floor of the pens
– correct double full stop
Response 11: Yes, accepted and revised. The floor of the pens was covered with sand-rich earth.
Point 12: 120 – please state clearly what the daily feed intake was by the end of the trial period with s.d. and what type of feed was given in that period??
Response 12: Yes, accepted and revised. The MF ration was fed to all donkeys prior to the start of the experiment.
The roughages consisted of mainly rice straw and alfalfa hay, which were chopped into approximately 1 cm lengths using a hay cutter. The concentrate feed was in crumbled feed form, and consisted of corn, soybean meal, cottonseed meal and a vitamin-mineral premix.
When feeding each time, first wet the roughage with water to make the water content of the roughage reach about 30%, and then mix the roughage and concentrated feed evenly before feeding
Point 13: 123 Major inconsistency here…… In discussion you mention that donkeys had ‘add libitum’ access ----- Please ADD very clearly here that the donkeys had add libitum access to the ration if that was the case – however, – if that was the case then why are there ‘3 meals per day’ at equal amounts?? If add libitum there should not be ‘meals’ …. Clarify this please or amend.
Response 13: Sorry, there are description mistake in English. Daily feed intake was measured to determine the fixed amount of experimental rations during the subsequent feeding trial. Once the experiment started, 1.2 times amount of experimental ration were divided into three equal meals and actually given at 8:30, 13:30, and 19:00.
Point 14: 132 Add libitum? Why not give more for the night period – the amount given in the evening would mean a large time of fasting which is not good welfare? What bedding were the donkeys on – did you observe/measure bedding eating?
Response 14: As we corrected in the above response. The amount of experimental rations were given more than the actual ingested for ad libitum intake. After evening feeding, there were still some ration left in trough for continuous eating at night. There are no extra feed on bedding
Point 15: Table 1. Feed – please state particle length of the forages and how they are presented Please explain how nutrient levels were measured – Wet Chemistry – Van Soest – How did you derive the amount of NFC – What is the purpose of NDF/NFC ratio? – it is never mentioned again in results….
Response 15: Now we have added particle size of forage in the Table 1. Nutrient level was measured with wet chemistry as stated in Chemical analysis. Since NFC was not mentioned in subsequent paragraphs. NFC and NDF/NFC ratio have been deleted now.
Point 16: Sample Collection
Move paragraph starting in line 148 above the preceding paragraph
Merge First paragraph under Sample Collection with Third paragraph and remove
repetitions
Response 16: The whole section has been revised and shorten.
Point 17: 185 – The term apparent total tract digestibility is confusing – normally reported as apparent digestibility co-efficient (%) and often abbreviated as d(%)…. But it is up to you to use this if you like….
More common is DMd (%) – or you could add (ad) for apparent digestibility = DMad(%),
NDFad (%) etc…
However – need to move the ‘unit’ from end next to your abbreviation
So should read ATTD (%) = ……….. x 100 at the very least?
Or can be given in (g/kg) = then you need to multiply by 1000
Response 17: The equation has been deleted now.
Results
Point 18: 210 – paragraph: could we please have a line in Table 2 highlighting amount of DM/day offered for comparison with actual intake??
Response 18: Yes, accepted and added.
Point 19: 215 – Table 2 – if truly ad libitum this should be defined as Voluntary Dry Matter Intake VDMI and could you please give this also as % of kg BW for comparison with previous papers in the discussion I am not a bit ‘fan’ of SEM – as it does not allow you to see variability differences between different diets – it is a misleading overall measure ok for statistics but for in depth analysis of the biological value of variation between diets the reporting of s.d. or s.e. per diet is much more valuable
Why are units in table now not ATTD? perhaps should read DMd; CPd; etc…
Response 19: As we noted in the above response. The amount of experimental rations was given more than the actual ingested for ad libitum intake. We have mentioned ration allowance in the material and method. Following you suggestion, Results in Table 2-5 have have revised and presented as means +/- standard errors.
Point 20: 223-225 – it is unnecessary to report p-values (p is larger than) if something is not significant – you have stated your sig. level in Methods. I would also suggest that from the table all you need to do is show the sig level between the highest and lowest – but this is
restating what is in table already and perhaps should be moved to discussion. –
Response 20: Great thanks for your nice suggestions. We have thoroughly revised the whole result description to make it as concisely as possible.
Point 21: 227-233 this is already discussion unless you give an association value – r2 - with this using individual Donkey results…..
Response 21: We have deleted the inappropriate description and thoroughly revised the whole result description to make it as concisely as possible.
Point 22: 230 - - again just report very shortly sig. differences as most of this text is clearly visible in the table so no need to report all – take out ‘p> …..
Response 22: Yes, accepted and revised now.
Point 23: 237 …. Start Table 3 title with capital letter – check editing throughout
Table 3 Measures presented here are unclear – possibly got lost in translation - perhaps
you need to replace some digestible with digestibility – please check and clarify (e.g. – what is the difference between Feed N and N Intake? What does Digestible N mean?
--- need to define this for the table but also in line 196- Define this in comparison to N
intake?)
Is this N Digestibility? - same for protein – is this CP? – if ‘predicted’ Digestible Protein need
to give the system used to predict this
Response 23: Thanks, all titles of tables have checked one by one to avoid those issues
Great thanks for your comments, Sorry, the description was indeed too vague and confusing. Now we have checked and revised the whole results in Table 3. Following N balance estimation per animal per day, the terms of daily feed intake, daily N losses in feces and urine were clearly mentioned, then the daily N retention were listed afterward. The terms of ‘digestible N or protein’ were deleted since digestibility results have been reported in Table 2. At the same time, the corresponding calculation and equation related to BV and NMR were checked and rephrased in Material and method section.
Point 24: 248-265 – again a lot of unnecessary repetition and description of table when this can be seen anyway – this needs to become more concise…. As per above
Response 24: Thanks, we have deleted the inappropriate description and thoroughly revised the whole result description to make it as concisely as possible.
Point 25: Line 267 – you mean GE Intake? here – again need to be clearer on Abbreviations GE in FE out
DE = DE of total feed? – in sentences below you talk about DE – but mean ATTE as defined in
methods? – perhaps need to define here as Feed ME?
ATTDE? Or DE/GE (%) would be consistent here = ATTDE
ME/DE (%) – could then be given as ATTME? --- this is not sig. whereas ME is??
Response 25: Great thanks, there are indeed inconsistent description. We have checked and revised the results in Table 5, and removed those confusing terms or result description to make them as correctly as possible.
Point 26: GE, the energy concentration of gross energy of the ingested feed; FE, Energy lost from feces per kilogram of feed. fecal energy; DE, Energy concentration of digestible energy in feeddigestible energy; ADE, apparent energy digestibility; UE, The energy lost from urine per kilogram of feed.urinary energy; ME, The energy concentration of feed metabolizable energy;
Response26: Yes, accepted and revised now. Following energy balance per animal per day, We have checked and revised the results in Table 5, and removed those confusing terms or result description to make them as correctly as possible.
Discussion Feed Intake
Point 27: 301 – nevertheless, somewhere need to clearly state how much was actually offered on a daily basis
Where there any left overs in the morning after the whole night?? If not then it is very likely that donkeys were eating bedding which may explain very low ADF and NDFd
Response 27: Yes, accepted and revised. Following your comments, we revised the related description about feed allowance and how feed intake was measured. All donkeys had free access to the MF ration (Table 1) prior to the start of the experiment. Daily feed intake was measured to determine the fixed amount of experimental rations during the subsequent feeding trial. The donkeys were housed with sand-rich earth beddings, and allowed to daily walk in outside pens with the same bedding. In the formal experiment, According to feed intake measurement prior to the start of the experiment, 1.2 times amount of experimental ration were divided into three equal meals and given at 8:30, 13:30, and 19:00 after the forage was wet with 30% water. Therefore, the donkeys can only eat feed left overs in the trough, and there are no any eating materials on the beding at night.
Point 28: 305 – yes but where is your underpinning evidence for this – no references for donkeys – horses for example, eat more concentrate feed because they have an ethological
requirement to carry out a certain amount of chewing behaviour in 24 hours – around 12-14
hours of ‘foraging’ in the wild – donkeys are very much the same spending most of daylight
and night on foraging behaviour 16-18 hours (Martin-Rosset, 2018; Mueller et al., 1994)!! – donkeys intake rates are ‘very slow’ compared to horses – but the feed you provided
would only keep them busy for around 8 hrs/24 hours, so surprised that this was ad libitum,
unless this specific feed takes them even longer to eat?
– the volume (in size/gut fill) is a strong factor in intake rates and amounts – see the
following papers – need to build on latest research in terms of feed intake behaviour –which is also valid for donkeys – also see Harris et al., 2017
Henneke, D.R. & Callaham, J.W. 2009. Ad Libitum Concentrate Intake in Horses, Journal of Equine Veterinary
Science, vol. 29, no. 5, pp. 425-427.
Argo, C. McG., Cox, J. E., Lockyer C. and Fuller Z. 2002. Adaptive changes in the appetite, growth and feeding
behaviour of pony mares offered ad libitum access to a complete diet in either a pelleted or chaff-based
form. Animal Science 2002, 74. pp. 517-528
Harris, P.A., Ellis, A.D., M.J.Fradinho, A.Jansson, V.Julliand, N.Luthersson, A.S.Santos and I. Vervuert. 2017.
Feeding conserved forage to horses: recent advances and recommendations; Animal (2017), 11:6, pp
958–967
Response 28: Yes, after a whole night, sometimes there will be a small amount of leftover material in the trough, sometimes it is very clean to eat, and there is no observation of the situation of eating bedding.
I have added relevant literature
Point 29: 308 where is the evidence that donkeys prefer short chopped forage? I think it is most likely the opposite….
Response 29: I did not find relevant literature, just found through observation, maybe the expression is not accurate. It has been observed in production that the donkey's roughage needs to be chopped shorter than cattle.
Point 30: 311 onwards – some very valid points but again need underpinning and exploring a bit more providing evidence from your results – maybe better as intro in next paragraph where the evidence is provided….
Response 30: I have added relevant literature
Point 31: 316 You write about metabolic diseases but need to give more detail (not necessarily here)– what diseases and are they related to a lack of fiber or an excess of sugar/starch in the diet??? Was your LF diet species specific?
Response 31: I have added relevant literature
Point 32: 317 4.4.2 – inconsistent title – before you refer in methods to ATTD – so why not use this here as a heading?
You then use the abbreviation D – for digestibility = not explained in methods and D not used in Table 2
Results repeated without values …. Scope for correlation here or discussion of % differences
instead e.g. NDFd decreased as fibre component of the diet decreased from HF to MF by X
% and from MF to LF by X % (p<0.01).
Response 32: The questions about ATTD and d have been revised before.
The duplicated parts have been deleted and modified according to your instructions.
Point 33: 322 – the ADFD was not lower for LF than MF group – mistake here….
Response 33: I have corrected this error
Point 34: 323 - 324 You then refer to pH – this is not explained as a measure in Methods and Results are not reported …. Either take out or add above…. No values are given anywhere Discussion weak and not backed up with data or literature You write ‘related’ - but there is no ‘relationship’ reported
Response 34: The content related to pH has been removed.
Some related literature has been added.
Point 35: 326 again does not make sense the DMD significantly reduces from LF to MF to HF ???
CPd – mostly repeat of results without values but not discussion - ??
Response 35: These pointless duplications have been deleted.
Point 36: 331 Organic matter or OM digestibility has not yet been mentioned before? And is not reported anywhere….
Response 36: The digestibility of organic matter has not been determined, so it will not be discussed here and deleted.
Point 37: 336 – should not report non significant results as decreases or increases – again no discussion here
Response 37: These pointless duplications have been deleted
Point 38: 339 Paragraph: In horses….. and explain dietary ratio / inclusion of starch – Follow up sentence ‘increase in starch therefore increases DMd….’ That is out of context and only valid up to a max. amount of starch in the diet which prevents too much overspill into the hindgut…
The conclusion is somewhat valid but could be explained better – possibly start with
conclusion and then use underpinning evidence – Discussion on passage rates very valid but
roughage is retained longer the less is eaten – the longer feed is retained the more starch will also be digested so a high concentrate diet generally is retained longer than a high forage diet in the same equid species !!
Response 38: Yes, accepted and revised.
Point 39: 350 need to be clear here – and refer to previous literature – not really referring to enough donkey literature and previous studies which are out there
Response 39: This section has been supplemented with literature
Point 40: 362 – disjointed – you talk about fiber fermentation and then jump to N excretion rates – what is the link? Mechanisms of increasing urinary N are not explored and unclear….
Response 40: I have deleted the fiber fermentation part。
Point 41: Mechanisms of increasing urinary N are not explored and unclear….
Response 41: Yes, accepted and revised.
Point 42: 371 – this is very basic – and only as long as high intake rates do not push soluble nutrients into the hindgut – could be explained better
Response 42: This section has been supplemented with literature
Point 43: 377 – indeed can you work out how long your donkeys were spending on feed intake time as if shorter than this their passage rates would have been quite slow – it also explains why when there was more ground sugar and starch component in the diet the feed intake went up in order to fulfil the natural feed intake behaviour ‘times’
Response 43: There is indeed no precise record of the time spent by the donkey for feeding, and the passage rates have not been determined.
Point 44: 380 – How do you know that CF is passing through the digestibility tract faster – need references – some of the explanations are very basic and you need to refer to health issues of the high SS diet.
Response 44: The CF-related content was deleted because it was unable to find a suitable literature support.
This section has been supplemented with literature
Point 45: The purpose of the study is unclear in the conclusion – is there a problem with donkeys being undernourished – on a welfare basis we have a much higher proportion of the diet as forage for equines (see Harris et al. 2016) and the latest research in equids and donkeys needs to be taken into account before giving such a low guideline advice – for example see Burden et al., 2009 (…gastric ulcers in donkeys fed a high sugar and starch diet) and Cox et al., 2007 (…. Increased risk of impaction colic when feeding extra / concentrated feed on top or instead of forages).
Response 45: The purpose of this study is to find a reasonable diet composition to ensure the healthy and rapid growth of donkeys, and to provide a reference and guidance for production practice. At the same time, enrich the data on donkey nutritional requirement through the research of this project.

Reviewer 2 Report
This manuscript gives valuable information regarding nutrition of donkeys. However, it needs major revisions before it can be accepted to be published. Although there are not many studies dealing with donkeys to be used as references by the authors, they can refer studies on horses, and deepen the discussion. However, I also suppose that there are more studies also on donkey nutrition. I think that a real discussion is now lacking. References are not given according to the instructions. Linguistic revisions are also needed.
Detailed comments:
- L21: donkey rising; perhaps husbandry or farming are better terms to be used instead of rising
- L26: TOTAL feces collection (not whole)
- L39: TOTAL tract digestibility
- L40: Decreasing (not decreasing)
- L42: please give p-values with the word "remarkably"
- L49-56: is not a common way to give the abbreviations here - not applied in Animals. Is not needed because the abbreviations are given also in the text.
- L65: working animal (?)
- L118: how many days per week they were outside. Were the animals inside on some days without any outdoor exercise?
- L127: is it a common way to feed concentrates in powder form? How you ensure chewing and saliva excretion?
- L131: what was the interval of this collection?
- L201 - : show the statistical model
- L214, Table 2: show the digestibilities in this Table only, and give the DMI with p-values in the text.
- L219: disappearance; this should be digestibility
- L222: extremely significant; this has to be changed "highly significant"
- L223-224: This information is already in table 2.
- L227: pairwise test? move this to Discussion
- L240: Head of Table 3 "Nitrogen balance ...". N retained has to be "Nretention"
- L243: "... LF group did not STATISTICALLY differ ..."
- L247: statistically significant
- L254: statistically significat
- L264: statistically significant
- L267: statistically significant
- L267-: the numeric values are not necessary to repeat in the text - they are alredy presented in the Table 4
- L271: head of Table 4: "Energy balance ..."
- Table 4: GE, DE, ME are energy INTAKES; FE and UE are energy CONTENTS; ADE is DIGESTIBILITY: thus improve the table to include corresponding subheadings and reorganize the items.
- L276: ad p-value
- L278-279: do not repeat the numeric values
- L280-281: no statistically significant
- L282-283: intermediate numerical ME/DE value
- L306: not statistically significant
- L311: fibriolytic (instead of fibrinolytic)
- L311-312: Reference is needed for this sentence.
- L314: is this study (Jouany) done with donkeys? Give the animal species.
- L316: reference is needed
- L325: reference is needed. Use results/references with horses if there are no studies with donkeys (effect of NDF)
- L326-329: no pH values were measured in this study. Thus, rewrite this part of the discussion and give relevant references (there are studies with horses)
- L330-: OMD was not studied in this study. There are many studies in horses were the effect of CP level and other intake levels are studied. Refere those studies.
- L336-338: This is just a result with no discussion. Is said already in Results
- L346-349: Discussion is missing. Use valid references, too.
- L356-367: there are several studies dealing with this in horses - use more references, not only this single one.
- L362: The main site of fiber digestion is large intestine. However, what is the meaning of this (L362-364) in this context?
- L365: discuss also about the effect the protein quality. It was lower in LF than in the other diets.
- L375: delete "for"
- L375-377: is this difference because you use only 2 references here, and it is known that various studies give time ranges at least up to 17 hrs for horses as well.
- L376-384: the retention times are not measured in this study. What is the meaning of this discussion/review of studies here? Remove?
- L394: change feeding practice to "practical feeding"
Author Response
- Response to Reviewer 2 Comments
- Thank you for your detailed review of my paper, you let me see the deficiencies of my article. Your careful and professional revisions have improved the quality of my articles, and I have learned a lot from them. The following is my point-to-point response to your comments.
- Detailed comments:
- Point 1: L21: donkey rising; perhaps husbandry or farming are better terms to be used instead of rising
Response 1: It has been replaced.
- Point 2: L26: TOTAL feces collection (not whole)
Response 2: Has replaced whole with TOTAL
- Point 3: L39: TOTAL tract digestibility
Response 3: Has replaced whole with TOTAL
- Point 4: L40: Decreasing (not decreasing)
Response 4: Has replaced decreasing with Decreasing
- Point 5: L42: please give p-values with the word "remarkably"(p<0.01)
Response 5: Has been added p-values
- Point 6: L49-56: is not a common way to give the abbreviations here - not applied in Animals. Is not needed because the abbreviations are given also in the text.
Response 6: This part has been deleted
- Point 7: L65: working animal (?)
Response 7: In the original literature, it was stated this way, which means that the animal is in service
- Point 8: L118: how many days per week they were outside. Were the animals inside on some days without any outdoor exercise?
Response 8: Weekly activities outside the house at 10h, about 1.5h per day
- Point 9: L127: is it a common way to feed concentrates in powder form? How you ensure chewing and saliva excretion?
Response 9: It is not accurate enough to say that it is a powder, it should be a cracked feed, crushing the concentrate into fragments.
- Point 10: L131: what was the interval of this collection?
Response 10: Feed samples are collected every half a month.
- Point 11: L201 - : show the statistical model
Response 11: The model was applied as follow:
Yijk = μ + Ti + Pj + Ck + eijk
where, Yijk was the response variable, μ was the overall mean, Ti was the fixed effect of the diet treatment (LF, MF, HF), Pj was the random effect of the period (j=1 to 3), Ck was the random effect of the animal (k=1 to 6) and eijk was the residual error.
- Point 12: L214, Table 2: show the digestibilities in this Table only, and give the DMI with p-values in the text.
Response 12: The form has been adjusted, and combined with the opinion of another review expert, DMI has been made separately.
- Point 13: L219: disappearance; this should be digestibility
Response 13: Has replaced disappearance with digestibility
- Point 14: L222: extremely significant; this has to be changed "highly significant"
Response 14: Has replaced extremely with highly
- Point 15: L223-224: This information is already in table 2.
Response 15: This part of duplicate content has been deleted.
- Point 16: L227: pairwise test? move this to Discussion
Response 16: Yes, accepted and revised.
- Point 17: L240: Head of Table 3 "Nitrogen balance ...". N retained has to be "Nretention"
Response 17: Has replaced N retained with Nretention.
- Point 18: L243: "... LF group did not STATISTICALLY differ ..."
Response 18: Yes, accepted and revised.
- Point 19: L247: statistically significant
Response 19: Yes, accepted and revised.
- Point 20: L254: statistically significat
Response 20: Yes, accepted and revised.
- Point 21: L264: statistically significant
Response 21: Yes, accepted and revised.
- Point 22: L267: statistically significant
Response 22: Yes, accepted and revised.
- Point 23: L267-: the numeric values are not necessary to repeat in the text - they are alredy presented in the Table 4
Response 23: This part of duplicate content has been deleted.
- Point 24: L271: head of Table 4: "Energy balance ..."
- Table 4: GE, DE, ME are energy INTAKES; FE and UE are energy CONTENTS; ADE is DIGESTIBILITY: thus improve the table to include corresponding subheadings and reorganize the items.
Response 24: It has been corrected according to your comments.
- Point 25: L276: ad p-value
Response 25: P-value has been added
- Point 26: L278-279: do not repeat the numeric values
Response 26: This part of duplicate content has been deleted.
- Point 27: L280-281: no statistically significant
Response 27: Statistically has been added in front of significant
- Point 28: L282-283: intermediate numerical ME/DE value
Response 28: Has added the word numerical
- Point 29: L306: not statistically significant
Response 29: Statistically has been added in front of significant
- Point 30: L311: fibriolytic (instead of fibrinolytic)
Response 30: Has replaced fibrinolytic with fibriolytic.
- Point 31: L311-312: Reference is needed for this sentence.
Response 31: The literature has been supplemented.
- Point 32: L314: is this study (Jouany) done with donkeys? Give the animal species.
Response 32: This article is about horses.
- Point 33: L316: reference is needed
Response 33: The literature has been supplemented.
- Point 34: L325: reference is needed. Use results/references with horses if there are no studies with donkeys (effect of NDF)
Response 34: The literature has been supplemented.
- Point 35: L326-329: no pH values were measured in this study. Thus, rewrite this part of the discussion and give relevant references (there are studies with horses)
Response 35: pH related content has been deleted and the literature has been supplemented.
- Point 36: L330-: OMD was not studied in this study. There are many studies in horses were the effect of CP level and other intake levels are studied. Refere those studies.
Response 36: This article does not determine OMD. OMD related content has been deleted and the literature has been supplemented.
- Point 37: L336-338: This is just a result with no discussion. Is said already in Results
Response 37: The discussion has been supplemented and the literature has been supplemented.
- Point 38: L346-349: Discussion is missing. Use valid references, too.
Response 38: The discussion has been supplemented and the literature has been supplemented.
- Point 39: L356-367: there are several studies dealing with this in horses - use more references, not only this single one.
Response 39: The literature has been supplemented.
- Point 40: L362: The main site of fiber digestion is large intestine. However, what is the meaning of this (L362-364) in this context?
Response 40: Yes, accepted and revised.
- Point 41: L365: discuss also about the effect the protein quality. It was lower in LF than in the other diets.
Response 41: Yes, accepted and revised.
- Point 42: L375: delete "for"
Response 42: Already delete "for"
- Point 43: L375-377: is this difference because you use only 2 references here, and it is known that various studies give time ranges at least up to 17 hrs for horses as well.
Response 43: Literature has been added.
- Point 44: L376-384: the retention times are not measured in this study. What is the meaning of this discussion/review of studies here? Remove?
Response 44: Retention times have been deleted
- Point 45: L394: change feeding practice to "practical feeding"
Response 45: Has replaced feeding practice with practical feeding.
You can also view attachments
